# Summer Nighttime Anomalies of Ionospheric Electron Content at Midlatitudes: Comparing Years of Low and High Solar Activities Using Observations and Tidal/Planetary Wave Features

**Yu Yin** , **Guillermo González-Casado \*** , **Adrià Rovira-Garcia** , **José Miguel Juan** , **Jaume Sanz** and **Yixie Shao**

Research Group of Astronomy and Geomatics (gAGE), Universitat Politècnica de Catalunya (UPC), Campus Nord, Edif. C3, C. Jordi Girona 1-3, E-08034 Barcelona, Spain; yu.yin@upc.edu (Y.Y.); adria.rovira@upc.edu (A.R.-G.); jose.miguel.juan@upc.edu (J.M.J.); jaume.sanz@upc.edu (J.S.); yixie.shao@estudiant.upc.edu (Y.S.)
\* Correspondence: guillermo.gonzalez@upc.edu

**Abstract:** In this study, midlatitude summer nighttime anomalies (MSNAs) are analyzed via observations and tidal/planetary wave features using measurements from the Formosat-3/Constellation Observing System for Meteorology, Ionosphere, and Climate (F3C) for 2007, a year with low solar activity, and 2014, a year with high solar activity. The total ionospheric electron content, $EC_{ion}$, an integrated quantity derived from F3C measurements, was used to compare the observational data. The $EC_{ion}$ values were derived from accurate radio-occultation-retrieved electron density profiles without assuming spherical symmetry and from a model that separated the ground total electron content into the plasmaspheric and the ionospheric electron content contributions. An analysis of the $EC_{ion}$ data set confirmed that MSNAs were present in three different regions of the world for the months surrounding the local summer solstice during both 2007 and 2014. In the southern hemisphere, the so-called Weddell Sea Anomaly showed a maximum increase in $EC_{ion}$, measured as the difference between nighttime and midday values, that was more than three times that in the northern MSNAs. For each individual MSNA, the corresponding maximum increases in electron content were similar between the two years analyzed, so they were not significantly affected by solar activity. Then, linear least-square fit to the frequency–wave number basis functions was used to derive the tidal and planetary wave components contributing to MSNAs. The main component that appears to produce the Weddell Sea Anomaly is D0, followed by SPW1, DW2, and DE1, in this order, which make secondary but still relevant contributions. The presence of MSNAs in the northern hemisphere was clearly supported by the migrating tide SW2 in combination with DE1. SW2 also supported an early morning MSNA being observed in the northern hemisphere. The main tidal and planetary wave signatures producing the MSNAs did not significantly differ between 2007 and 2014.

**Keywords:** ionospheric anomalies; WSA; MSNA; radio occultation; COSMIC/Formosat-3

## 1. Introduction

The term midlatitude summer nighttime anomaly (MSNA) refers to the electron density (ED) or total electron content (TEC) in Earth's ionosphere being greater during the night than during noon, which is a phase reversal compared with the usual diurnal cycle of the ionospheric ED or the TEC. MSNAs are known to occur in three regions [1–4], with all of them located within the middle magnetic latitudes. The first, discovered in the 1950s using ionosonde measurements [5], is commonly known as the Weddell Sea Anomaly (WSA), and is observed in the southern hemisphere around geographic longitude $-90°$. The two other regions, observed in the northern hemisphere, are identified simply as MSNAs [1–3], with no unified terminology to describe them. One is located near the Bering

Sea, around geographic longitude 150°, and in this study, we refer to this MSNA as the Bering Sea Anomaly (BSA). The second is observed in the North Atlantic region, around −45° geographic longitude, and is herein referred to as the North Atlantic Anomaly (NAA).

The most widely studied MSNA is the WSA in the southern hemisphere, not only because it was the first to be discovered but also because it is more prominent than the anomalies in the northern hemisphere. The WSA has been analyzed using TEC measurements from on-board satellites [6–8] as well as ground TEC measurements obtained from the Global Navigation Satellite System (GNSS) [7,9]. Additionally, in situ ED measurements taken by low Earth orbit satellites have been employed to investigate the MSNAs [3,10].

With the launch of the Formosat-3/Constellation Observing System for Meteorology, Ionosphere, and Climate (F3C) in 2006, a large data set of ionospheric ED profiles has been retrieved from radio occultation (RO) measurements, particularly at middle latitudes due to the inclination of the F3C satellite orbits. Using these RO-retrieved ED profiles from F3C, the characteristics of MSNAs at different altitudes have been analyzed [1,4,7,11]. Additionally, the TEC calculated by vertically integrating ED profiles from F3C has also been used as a proxy for the study of the MSNAs [4,11], since it provides a simple and direct representation of the ionospheric electron content (EC) variability without needing to limit a study to a fixed altitude. The EC derived from RO measurements is more suitable than the ground TEC in an analysis of MSNAs, since these anomalies are mostly located over seas or oceans where ground GNSS stations are scarce or absent. Hence, ground TECs provide limited spatial coverage for the study of MSNAs.

The formation of the WSA has been attributed to a combination of thermospheric neutral winds, solar photo-ionization, and magnetic declination effects [12]. The effective wind approximately peaks at the magnetic dip angle of 45° in both hemispheres [2]. Due to the magnetic declination configuration in the WSA (and the other MSNA regions), the magnetic dip angle turns out to be close to 45° at the specific geographic longitudes and magnetic latitudes where MSNAs are observed [2]. At this angle, the largest effective wind values give rise to a large amount of plasma drift along the magnetic field lines at these MSNA locations, which increases the amount of ionized material that comes from the plasma transported upward and equatorward from high latitudes during local summer. In fact, this transport leaves the plasma at higher altitudes, where recombination is lower, and for this reason, the plasma persists for a longer time during the nighttime. Several observational studies based on F3C electron density seem to support this interpretation [2,13,14]. Another feature that has been suggested to contribute to MSNA formation is the offset between the geomagnetic and geographic equators. This offset results in larger equatorial ionization anomaly crests towards higher geographic latitudes, which combined with the specific configuration of declination angles in the MSNA locations, gives rise to a stronger equatorward neutral wind, which supports the accumulation of ionized material at higher altitudes [13]. However, more recent analyses of the WSA and the BSA concluded that the previously mentioned mechanisms are not sufficient in explaining these MSNAs. Instead, these MSNAs have been suggested to be produced by longitudinal changes in the neutral winds and neutral densities attributed to the varying latitudinal distances from the energy input of the auroral zone [15,16].

Additionally, frequency–wave number analysis has been applied in the past decade to study the ionosphere [17–19] and, in particular, to analyze MSNAs [3,11,20]. The integrated ionospheric EC from F3C RO-retrieved ED profiles has been used to investigate the tidal and planetary wave (T/PW) signatures contributing to the formation of the WSA [11]. The purposes of this approach and other related studies have been to understand the origin of MSNA in the context of ionosphere–atmosphere coupling through upward-propagating tides that are excited in the lower and middle atmospheres or, alternatively, by means of the in situ thermosphere generation of ionospheric tidal features [17,19,21,22]. Hence, an approach based on a frequency–wave number analysis of the ionosphere is important for supporting and complementing investigations based on model simulations and for gaining insight into the atmospheric/thermospheric mechanisms responsible for the formation of

MSNAs. To this end, the use of a reliable observational quantification of the ionospheric EC is of great importance. The dynamics and climatology of the ionosphere are highly variable, and in particular, the size of the ionospheric region can be subject to significant changes. Consequently, the separation of the ionosphere from the plasmasphere must be taken into account [23–25] to properly analyze MSNAs in the ionospheric region. However, in previous studies based on integrated ED profiles from ROs, the customary approach has been to calculate the ionospheric EC from the altitude of the F3C satellite orbits (approximately 800 km) down to lower altitudes [4,11]. Similarly, using measurements from on-board F3C antennas, the plasmaspheric electron content has been estimated as the TEC over the satellite orbit and subsequently used to analyze MSNAs in the plasmasphere [8]. These approaches do not take into account that the relative EC contributions from the ionosphere and plasmasphere change not only with latitude but throughout a day, throughout a year, and with the solar cycle period [23,24,26,27]. Hence, to focus the study of MSNAs on just the ionospheric EC, a method that adequately separates the ionospheric from the plasmaspheric contributions to the ground TEC in RO-retrieved ED profiles should be applied [26,28]. In addition, the ED profiles used in previous studies were products from the F3C Data Analysis and Archive Center, which were based on the assumption of spherical symmetry in the ED distribution. However, more precise methods of deriving the ED profiles from ROs that do not rely on a spherical symmetry assumption are currently available [26,29].

In this study, an empirical analysis of the MSNAs is performed using accurate ED profiles retrieved from F3C ROs without assuming spherical symmetry, and by removing the plasmaspheric contribution to the TEC to specifically analyze the EC from the ionosphere region, $EC_{ion}$ [25,26]. The goal of this study is to demonstrate that this variable can be used to reliably quantify the observational features of MSNAs and their main T/PW signatures. Toward this aim, increases in nighttime EC and the T/PW signatures of the different MSNA regions are compared based on $EC_{ion}$ data for two years with extreme solar cycle activities: 2007, specifically, during the beginning of the solar minimum between solar cycles 23 and 24; and 2014, specifically during the maximum of solar cycle 24. According to the 10.7 cm solar radio flux indicator, $F_{10.7}$, during 2007, the solar activity was low and close to the minimum, with a monthly average of $F_{10.7}$ ranging from 65 to 85 sfu and the monthly mean sunspot number never exceeding 30. In contrast, during 2014, the maximum of solar cycle 24 occurred, with a monthly average value of the $F_{10.7}$ indicator always greater than 120 sfu and reaching a peak of around 170 sfu. The monthly mean sunspot number was greater than 90 during 2014 and reached a maximum close to 150.

To our knowledge, studies covering years with both high and low solar activities and focused on comparing the T/PW components that produce MSNAs during both periods, jointly quantifying the main observational features of anomalies in both hemispheres, are lacking. This type of comparison should be of great interest in reinforcing and extending the results from previous analyses mostly based on low solar activity periods, eventually providing new hints and constraints relevant to theoretical models that explain the origin and characteristics of the MSNAs.

The manuscript is organized as follows. In Section 2, we summarize the methodology used to obtain the reference measurements used in the observational study and to subsequently derive the T/PW components contributing to MSNAs. Section 3 is devoted to presenting and describing the results and is separated into two parts. In Section 3.1, we quantify and compare the magnitude of the MSNAs in the different regions and between two years with low and high solar activities. Section 3.2 is devoted to studying the main T/PW signatures contributing to the MSNAs in 2007 and 2014 using a linear least-square model fitted to frequency–wave number basis functions. A discussion of the results of this study compared with those of previous studies is presented in Section 4. Finally, the main conclusions are summarized in Section 5.

## 2. Methodology

### 2.1. Ionospheric Total Electron Content from F3C ROs

The methodology used to determine the specific contribution of the ionospheric region to the ground TEC at a given location using RO measurements was developed and tested in several previous studies [25,26,28,29]. The main aspects of this method, which follows two steps, are highlighted below. More details can be found in the cited references.

In the first step, ED profiles are retrieved from RO measurements using an improved Abel inversion technique known as the separability method (SM) (see [26] and the references therein). The SM is a TEC-aided inversion technique that takes into account the horizontal density gradients by means of external TEC measurements provided by accurate global ionospheric maps (GIMs). The SM improves the accuracy of the ED profiles by more than 30–40% with regard to retrievals from the classic Abel inversion based on the spherical symmetry assumption [29], especially in the bottom-side ionosphere region. In fact, the SM can achieve F2-layer peak electron density values with an accuracy similar to the collocated ionosonde measurements [29]. In the SM, the ED at a given longitude $\lambda$, latitude $\phi$ and height $h$, $N_e(\lambda, \phi, h)$, is expressed as the product of two functions: a shape function, $S(h)$, representing the vertical variation in ED, and $TEC(\lambda, \phi)$, the external TEC measurements used to describe the horizontal ED variations in the region where RO measurements have been obtained. In this study, the TEC measurements are derived from the GIMs provided by the International GNSS Service, specifically the rapid GIM products available at https://cddis.nasa.gov/archive/gnss/products/ionex/, accessed on 7 February 2022. Then, using the TEC values from the GIMs, the solution for the $S(h)$ profile is retrieved from each RO following the procedure described in [26,29].

Subsequently, in the second step, the topside ionospheric and bottom-side plasmaspheric (TIP) portions of the $S(h)$ profile are fitted to a model function that considers two main components: one representing the variation in altitude of the density of the $O^+$ ion population, the main topside-ionospheric constituent, and one representing the altitude variation in the $H^+$ ion population dominating the plasmasphere [25,26,28]. The altitudes are used to fit the model range from a fixed height, taken at a few kilometers over the F2-layer peak altitude, to approximately the altitude of the F3C satellite orbit. The topside ionospheric component of the model is described by an exponential function decreasing with altitude according to a vertical scale height, $h_s$, which is a free parameter [25,26]. On the other hand, the plasmaspheric component is described by the chemical equilibrium plus diffusion equilibrium $H^+$ model (CPDH) introduced in [28], which is a simplification of the original model described in [30]. Thus, the model fit function for the TIP region of the ED profile, $S_{fit}(h)$, can be expressed as follows:

$$S_{fit}(h) = Ae^{-h/h_s} + S_{H^+}(h),  \tag{1}$$

where $A$ and $h_s$ are constant free parameters used to describe the variation in altitude for the topside-ionosphere component, while $S_{H^+}(h)$ is the CPDH model function described by Equation (4.6) of [28]. The accuracy of Equation (1) in reproducing RO-retrieved ED profiles has been demonstrated to be significantly better than that achieved with conventional models using a single component [25,28]. Originally, the plasmaspheric component was modeled by a constant term [25,26], but the use of the more sophisticated CPDH model in Equation (1) has been shown to improve the accuracy of the fits by approximately 40–50% compared with the original constant-term model [28].

After model fitting of the TIP part of the ED profile, the original RO-retrieved ED profile in the bottom-side ionosphere region and around the F2-peak altitudes is continued towards higher altitudes using only the ionospheric component of the model Equation (1), that is, using the ED derived from the exponential term in that equation. Then, the resulting profile is integrated over the altitude to obtain the TEC from the ionosphere

region, $EC_{ion}$. This calculation is described in detail in [26] and is performed according to the following equation:

$$EC_{ion}(\lambda, \phi) = TEC(\lambda, \phi) \left[ \int_{h_0}^{h_{fit}} S(h)dh + h_s A \exp\left(-h_{fit}/h_s\right) \right], \tag{2}$$

where $h_0$ is the minimum altitude sampled by the RO measurements, which must be in the range from 80 to 150 km, while $h_{fit}$ is the minimum altitude used to perform the TIP model fitting. In Equation (2), the first term between brackets multiplied by the TEC is the integral over the altitude of the original ED profile, while the second term between the brackets is obtained after integrating the extrapolated ED from the ionosphere component in Equation (1), providing the topside-ionospheric contribution to $EC_{ion}$. Thus, the resulting $EC_{ion}$ does not include the contribution from the plasmasphere. The contribution of the plasmaspheric EC to the ground TEC, $EC_{pl}$, can be derived simply from the difference $TEC(\lambda, \phi) - EC_{ion}(\lambda, \phi)$. For this study, after performing least-square model fitting for the TIP part of ED profiles, only fits with mean relative errors (model versus observations) less than 10% were selected for the derivation of $EC_{ion}$ data to ensure that the model fits were sufficiently accurate.

We processed the F3C RO measurements following the previous two-step methodology for two years: 2007 and 2014. The RO measurements from all days during 2014 were processed. However, for 2007, only 50% of the days, evenly spaced throughout the year, were considered to avoid a possible influence in the results caused by a data sample with a very different size for high and low solar activity periods. Note that many more ROs were available in 2007 than in 2014 due to the decay of the orbit of several F3C satellites since the constellation was launched in 2006, largely affecting the number of available ROs during 2014. In any case, the resulting data set was verified to have a sufficient coverage of the geographic longitudes, magnetic latitudes, months of the year and local times (LTs) of interest for this study. The total number of data obtained was approximately equal to $2.2 \times 10^5$ and $2.5 \times 10^5$ for 2007 and 2014, respectively.

To highlight the differences between the ionospheric EC and the TEC, in Figure 1 we present the LT variation in the mean values of TEC, $EC_{ion}$, and $EC_{pl}$ over the approximate region covered by the WSA during January and the BSA during June in the two years analyzed, 2007 and 2014. For the WSA (left column of Figure 1), the total contribution of $EC_{pl}$ (dashed lines in Figure 1) to the ground TEC (blue lines) can be seen as being far from negligible during nighttime, when it reached up to 20–25% in 2007 and was slightly lower, around 15%, in 2014, the year of the solar maximum. In the case of the BSA (right column of Figure 1), the proportion of $EC_{pl}$ even exceeds 30%. One can observe the clear enhancement in the ionospheric electron content during night LTs (solid lines in Figure 1), while the plasmaspheric electron content shows little variation and no enhancement during the night. Similar results are achieved for the northern MSNA during local summer periods (right column of Figure 1). Thus, one can conclude that the $EC_{ion}$ data are able to clearly predict MSNAs in the ionosphere region, regardless of the magnitude of the solar activity.

On the other hand, taking for example the plot for January 2007 from Figure 1, the percentage of relative increases in the nighttime EC with regard to midday values derived using $EC_{ion}$ is more than 65%, while the corresponding relative increase calculated from TEC (including both $EC_{ion}$ and $EC_{pl}$ contributions) is only around 45%. In the other cases from Figure 1, the relative increase in TEC is always smaller than the one from $EC_{ion}$. Hence, using TEC instead of $EC_{ion}$ leads to an underestimate of the magnitude of the MSNAs when measured with respect to the midday period, which could give rise to a poor quantification and even difficulty detecting MSNAs, particularly in the case of the weakest MSNAs in the northern hemisphere.

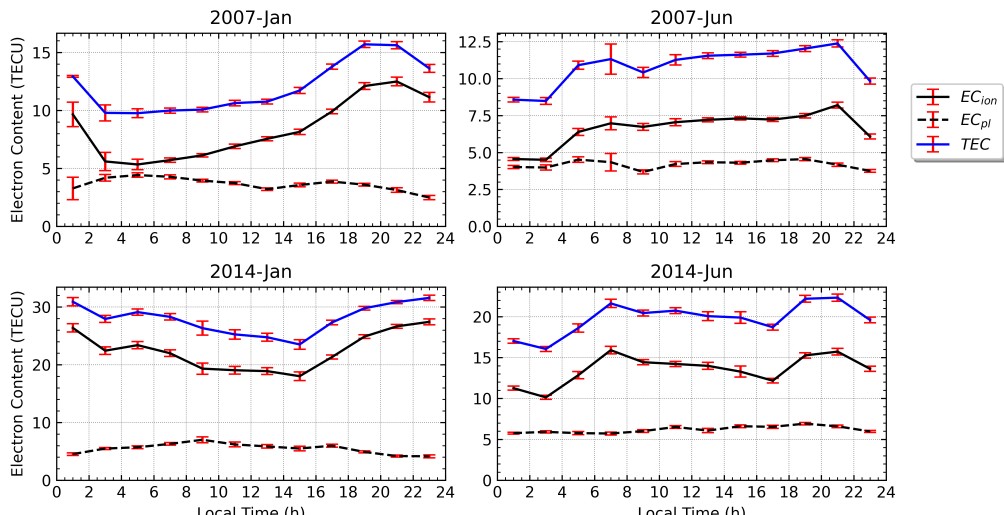

**Figure 1.** Variation in the average TEC, $EC_{ion}$, and $EC_{pl}$ with the LT during local summer over the approximate regions covered by the WSA (geographic longitude $[-150°, -60°]$ and magnetic latitude $[-65°, -40°]$) and the BSA (geographic longitude $[120°, 180°]$ and magnetic latitude $[40°, 65°]$). The corresponding months are January and June in 2007 and 2014, respectively. The standard error of the mean (red bar) was estimated using the sample standard deviation divided by the square root of the number of measurements in each bin.

### 2.2. Modeling Constituent Tidal and Planetary Wave Components of Ionospheric MSNAs

In the present study, based on the observational data sample of $EC_{ion}$ and the method from [11], the frequency–wave number analysis was applied to determine the main T/PW ionospheric signatures responsible for the MSNAs during 2007 and 2014. For this aim, $EC_{ion}$ values were binned at $5°$ intervals of magnetic apex latitude [31] in the midlatitude ranges from $40°$ to $65°$ of the northern and southern hemispheres and for each month of the two years. Then, to describe the tidal and stationary planetary wave components of the ionosphere [11,19], the following set of basis functions was applied:

$$F(\lambda, t) = \overline{F} + \sum_{n=1}^{3} \sum_{s=-4}^{4} \hat{F}_{n,s} \cos\left(n\Omega t - s\lambda - \hat{\Psi}_{n,s}\right) + \sum_{s=1}^{4} \hat{F}_s \cos\left(s\lambda - \hat{\Psi}_s\right), \tag{3}$$

where $\overline{F}$ is the zonal time mean, which is estimated from the time averaged $EC_{ion}$ over all longitudes in a given month and apex latitude bin. $\hat{F}_{n,s}$ and $\hat{\Psi}_{n,s}$ are the amplitude and phase of the tidal waves, respectively, while $\hat{F}_s$ and $\hat{\Psi}_s$ are the corresponding amplitude and phase of the planetary waves, respectively. $t$ is the universal time, $\Omega = \frac{2\pi}{24}$ is in units of inverse hour, and $\lambda$ is the geographic longitude. Finally, using the linear least squares method, the monthly data in each magnetic apex latitude bin were fit to the basis functions.

After obtaining the fitting results, the stationary planetary wave components 1 to 4 (last term in Equation (3)) and diurnal, semidiurnal, and terdiurnal tidal components ($n$ from 1 to 3, respectively, in the second term of Equation (3)), with zonal wave numbers $s = -4, \ldots, +4$ (eastward for positive $s$) were extracted for each magnetic apex latitude bin and month. Using the set of basis functions from Equation (3), the best fits to $EC_{ion}$ data were achieved with 10–15% mean relative errors for the two years analyzed in the months surrounding the June and December solstices, the periods when the MSNAs were more prominent. In the worst-case scenario, the mean post-fit relative error for some specific latitude bins reached 20%. The absolute error of the best-fit models with regard to observations was typically 0.5–1 TECU in 2007 but increased to 2–3 TECU in 2014. Recall that this result is consistent with the ionosphere EC values for the solar maximum being more than two times larger than that for periods with low solar activity, as illustrated in Figure 1.

In the following, we use common acronyms to identify the electron content contribution from different T/PW components in Equation (3). For the tidal components, the first letters D, S, and T indicate the diurnal, semidiurnal, and terdiurnal components, respectively, with *n* being from 1 to 3. The second letters E or W are used to identify tidal waves moving longitudinally eastward (wave number $s > 0$) or westward (wave number $s < 0$), respectively, and the final number indicates the wave number. For the special case of standing waves, $s = 0$, the notation is D0, S0, and T0. Different from the tidal components, SPW represents the stationary planetary waves, followed by an integer from 1 to 4, which gives the number of longitudinal maxima for a wave. Since Equation (3) depends on longitude and time, the model results are presented using time-dependent curves for fixed longitudes. SPWs have a constant value at a given longitude, so they are represented as horizontal lines when displaying their LT variation. On the other hand, tidal waves for a fixed longitude are represented as a cosine function along the LT axis with the amplitude and phase values as obtained after the best fit to Equation (3) and with a period that depends on the value of *n*.

## 3. Results

### 3.1. Quantifying the MSNA from F3C RO Observations

A first qualitative look at the approximate location and relevance of the different MSNAs in the two years analyzed is presented in Figure 2. The color scale in the maps represents the percentage of positive EC increases measured by the relative difference in $EC_{ion}$ between some nighttime and near midday LTs selected for illustrative purposes. For the northern anomalies in the top row of Figure 2, the relative difference in $EC_{ion}$ at 21:00 compared with 13:00 LT is shown, while for the WSA in the southern hemisphere, the relative difference in $EC_{ion}$ at 0:00 compared with 13:00 LT is shown in the bottom row of Figure 2. The colors in the maps correspond to values averaged over the two months when the anomalies are more prominent, according to our data set (June and July for the northern hemisphere and December and January for the southern hemisphere). One can also observe black solid curves crossing the maps in Figure 2, which correspond to lines of constant magnetic apex latitude [31].

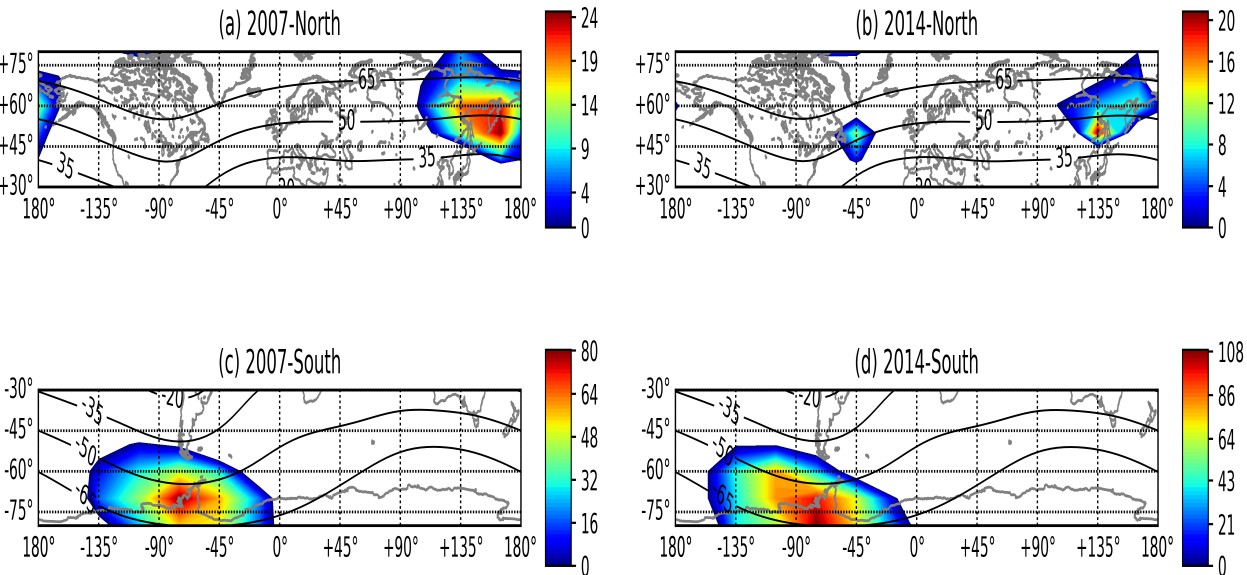

**Figure 2.** Percentage of relative difference between $EC_{ion}$ values during nighttime and midday LTs. (**a**,**b**) For 2007 and 2014, respectively, the relative differences between $EC_{ion}$ at 21:00 and 13:00 LT in the northern hemisphere using data from June and July. (**c**,**d**) For 2007 and 2014, respectively, the relative differences between $EC_{ion}$ at 0:00 and 13:00 LT in the southern hemisphere using data from January and December.

Figure 2 (bottom row) shows the presence of the well-known WSA in the southern hemisphere during 2007 and 2014. Clearly, the WSA affects a larger geographic region than the MSNAs in the northern hemisphere. Moreover, the WSA shows a larger increase in electron content during the nighttime compared with the BSA and the NAA for both years. In fact, in 2007, only the BSA is observed in the northern hemisphere, indicating that the NAA is the weakest MSNA, although, as we will see later, it can still be detected during the evening in 2007.

This study has systematically analyzed and quantified the observed magnitude of the different MSNAs for the magnetic apex latitude interval $[40°, 65°]$ in the northern and southern hemispheres, where the main increases in ED produced by MSNAs have been identified in previous studies [1,3,7,10,11], being also consistent with a preliminary inspection of our data set. To compare the different MSNA regions and different periods of time, we considered the $EC_{ion}$ values in the aforementioned magnetic apex latitude intervals to calculate the relative difference, $D$, in $EC_{ion}$ at different LTs throughout the day in comparison with that at midday. The calculation was performed per month and per given LT and longitude bins, where the bins for geographic longitude were selected to cover regions where the MSNAs are typically observed, specifically within the intervals $[-150°, -60°]$ for the WSA, $[-60°, 0°]$ for the NAA, and $[120°, 180°]$ for the BSA. Thus, the $EC_{ion}$ data sample was binned in intervals of $30°$ for longitude and two hours of LT for each month of the year. Then, the monthly mean $EC_{ion}$ value for each bin, $EC_{ion}(\tau, \lambda)$, was calculated, where $\tau$ and $\lambda$ are, respectively, the central values of the LT and longitude intervals of the bin. On the other hand, for each longitude interval, the maximum monthly mean $EC_{ion}$ value found in the bins covering the midday period, from 11:00 to 15:00 LT, $EC_{ion}^{md}(\lambda)$, was subsequently determined. Finally, $D$ was calculated from the following relative difference:

$$D(\tau, \lambda) = \frac{EC_{ion}(\tau, \lambda) - EC_{ion}^{md}(\lambda)}{EC_{ion}^{md}(\lambda)}. \qquad (4)$$

Note that we do not consider the reference value $EC_{ion}^{md}$ at a fixed LT, for example, at 12:00. Instead, we adopt a more conservative approach, considering the maximum electron content observed during the middle of the daylight hours because at other locations where the MSNAs are not observed, for example, other midlatitude or equatorial regions, the peak EC during the day may not occur exactly at midday but a few hours afterwards [26,28]. Note also that $D$ is a measure of the magnitude of each MSNA averaged over the full interval of magnetic apex latitudes considered.

In the case that no data or just one value is found in a given bin, the value assigned to that bin was calculated by interpolating the values from surrounding bins, that is, a two-dimensional interpolation in LT and longitude. The interpolation was performed only in two cases for 2007, for the bins centered in 2:00 LT in January and 4:00 LT in December for the longitude interval at $-75°$. In 2014, only one interpolation was performed in December for 2:00 LT and longitude $-75°$. Thus, the interpolations were constrained to a few specific locations and times related to the WSA.

The LT variation in the parameter $D$, in %, showing the EC increase or decrease observed in comparison with the midday hours, is presented in Figures 3–5, for the WSA, BSA, and NAA, respectively, and for the months when the anomalies were more prominent, that is, when the largest increases in nighttime EC according to the parameter $D$ were measured. These months were December and January for the WSA and June and July for the northern MSNAs. The different curves in these figures correspond to different longitude intervals, shown by the labels in the figures. Moreover, in the different regions covered by the MSNAs, the maximum nighttime EC increase measured by the parameter $D$ was derived during the periods surrounding the summer solstices in the northern (from May to August) and southern (from November to February) hemispheres, respectively. The resulting maximum values, $D_{max}$, for each of the three MSNA regions in the two years analyzed are presented in Table 1, also indicating the month, the longitude, and the LT where the maximum percentage of EC enhancement was obtained for each case. The range

of LTs considered to search for the $D_{max}$ values in Table 1 was from 18:00 to 8:00, including not only nighttime but also evenings and early morning.

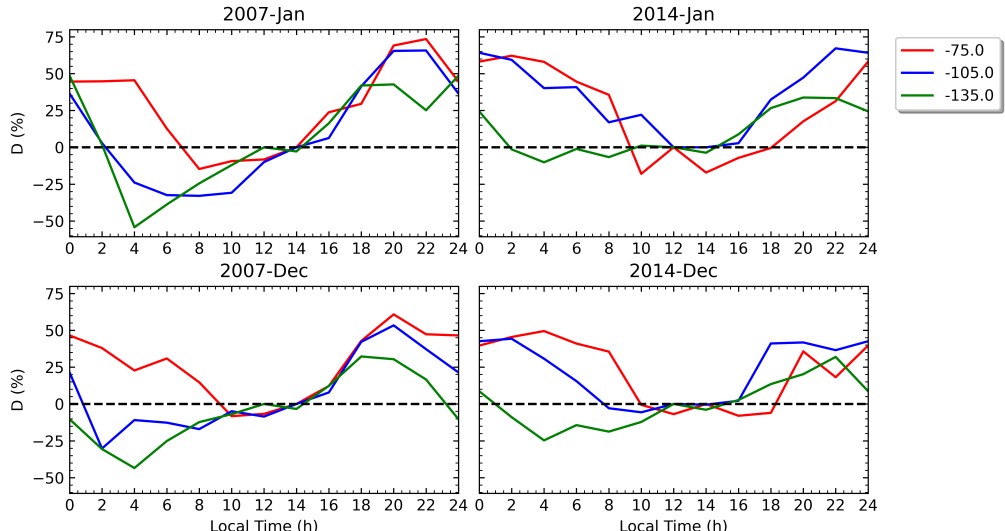

**Figure 3.** Monthly averaged LT profile of parameter $D$ for different ranges of longitudes covering the WSA (bin size of 30° for the values indicated in the top right corner of the plot). The results correspond to the two months when the anomaly is more prominent during (**left** column) 2007 and (**right** column) 2014.

**Table 1.** Maximum percentage of nighttime electron content enhancement with regard to the midday period in each MSNA for 2007 and 2014.

| MSNA | Year | Month | LT (h) | LON (deg) | $D_{max}$ |
|------|------|-------|--------|-----------|-----------|
| WSA | 2007 | January | 22:00 | −75° | 73% |
| BSA | 2007 | July | 20:00 | 165° | 17% |
| NAA | 2007 | July | 18:00 | −45° | 9% |
| WSA | 2014 | January | 22:00 | −105° | 67% |
| BSA | 2014 | June | 8:00 | 165° | 21% |
| NAA | 2014 | June | 20:00 | −45° | 21% |

Figure 3 shows the presence of the well-known WSA in the southern hemisphere, which can be clearly appreciated in both years during the two months presented in the figure. In fact, the WSA can also be observed during November and February for longitudes between −60° and nearly −150°, with December and January being the months when the anomaly is more prominent. During those months, the WSA can be seen between the evening and nearly 8:00 LT in 2007 and 2014, with positive or around zero values of $D$ measured at some longitudes examined in Figure 3. The WSA clearly lasts longer than the MSNAs in the northern hemisphere, as seen by comparing Figure 3 with Figures 4 and 5. Moreover, the WSA shows larger increases in EC compared with that of the BSA and the NAA for both years. In fact, the most striking difference between the three MSNA regions, as clearly seen in Table 1, is that the WSA presents maximum increases of around 70% while, in the northern hemisphere, the BSA and NAA reach only maximum increases of around 20%, more than three times smaller.

An important remark about the MSNAs in the northern hemisphere is that, rather than being nighttime anomalies, they could be better termed as evening/early night anomalies since they are observed around or shortly after sunset LT, specifically from 18:00 to 22:00. Moreover, Figure 4 shows the presence of a second relevant peak of $D$ during early morning, 6:00–8:00 LT, when the BSA can also be observed. In fact, the BSA reaches its maximum increase in EC at 8:00 LT in 2014 and at 20:00 LT in 2007 (see Table 1).

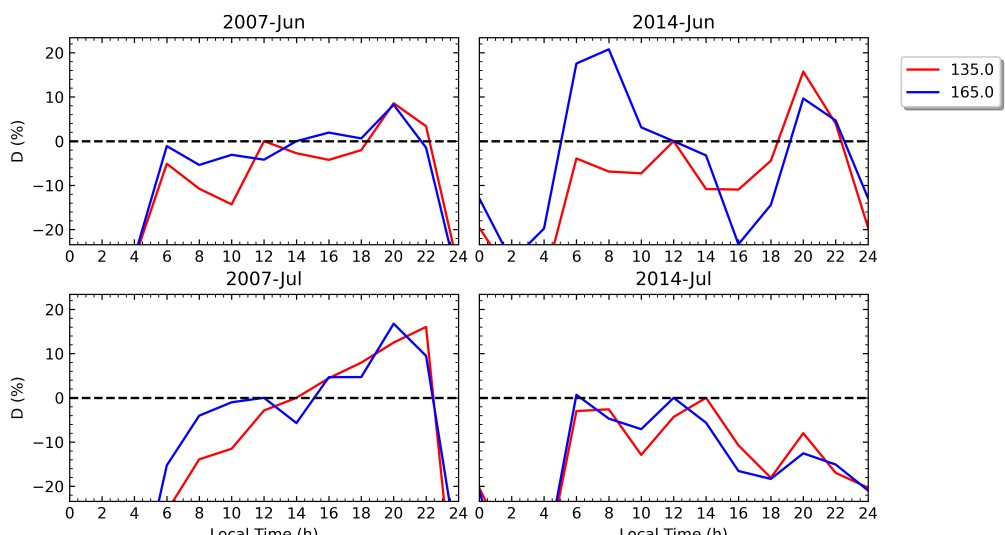

**Figure 4.** The same as Figure 3 but for the longitudes covering the BSA in June (**top** row) and July (**bottom** row).

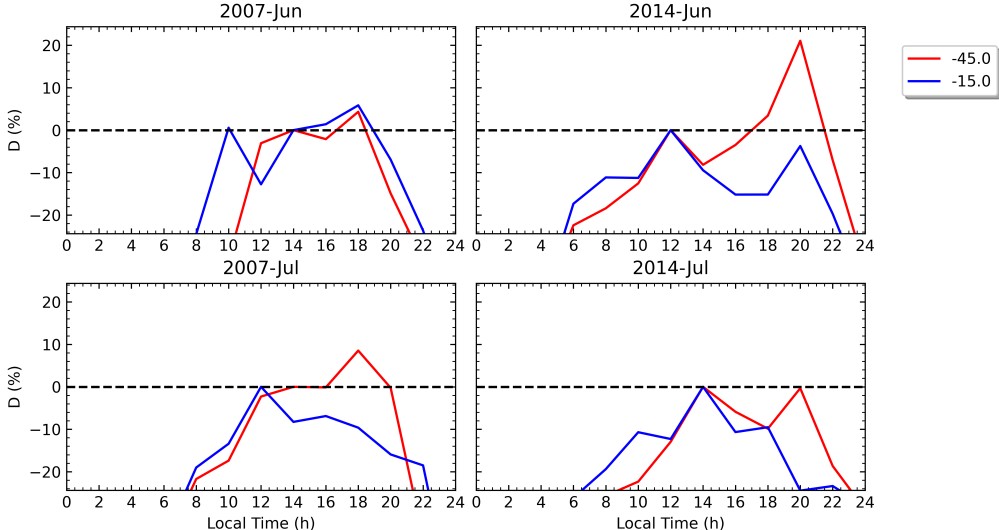

**Figure 5.** The same as Figure 4 but for the longitudes covering the NAA.

Regarding the WSA, Figure 3 shows that the rate of growth in parameter $D$ is greater in the eastern sector (red lines) than in the western sectors (blue and green lines) during the night. Then, after midnight, the WSA clearly becomes more prominent in the eastern than in the western areas in 2007 and 2014. This provides evidence of an eastward drift in the LT frame of the WSA, which has been reported in previous studies for years of low or minimal solar activity [3,11,21]. This eastward drift is also observed during the solar maximum in 2014 but with some variations with regard to 2007. In particular, the onset of the WSA in the eastern longitudes seems to occur earlier in 2007 than in 2014, and the maximum values attained by $D$ for those longitudes are clearly achieved earlier in 2007 than in 2014. This suggests that the WSA develops at an earlier LT during a period of low solar activity than during a year with a solar maximum.

Due to the different trends observed in $D$, the nighttime EC increases in the WSA region can appear larger in 2007 or 2014 depending on the longitudes and LTs considered. Some previous studies have suggested that MSNAs in general are strengthened by solar activity and are more prominent during periods surrounding a peak in the solar cycle [8,10,16]. However, the fact is that the maximum percentage of EC enhancement for

the WSA, without taking into account the time of year, the longitude, and the nighttime LT, is similar between 2007 and 2014, around 70%, and occurs in January at 22:00 LT, as shown in Table 1 and Figure 3. Thus, one can conclude that the solar activity has no relevant influence on the $D_{max}$ value reached by the WSA. Nevertheless, the specific variability of the WSA for a given range of longitudes shows some differences in the two years analyzed, indicating that part of the behavior of the WSA is affected in some way by the solar activity.

In the case of the BSA and the NAA, one can reach conclusions similar to those of the WSA concerning the similarity of their maximum magnitude between 2007 and 2014 as measured using $D_{max}$ in Table 1 (see also Figures 4 and 5). According to the results presented in that table, in the northern hemisphere, the maximum increases in nighttime EC are similar between the BSA and the NAA and between the two years analyzed for both regions, with values of around 20%, except for the NAA in 2007, which shows slightly lower values of $D_{max}$, close to 10%. However, Figures 4 and 5 also illustrate the differences in the LT variations in parameter $D$ for the two years analyzed, pointing again towards some influence of solar activity on the behavior of the northern MSNAs. In particular, for 2007, the BSA and NAA show the largest $D$ values in July, while for 2014, they are observed in June. In fact, in July 2014, $D$ never achieves positive values during the night for both anomalies (bottom right panels in Figures 4 and 5). On the other hand, the early morning peak of $D$ shown by the BSA reaches large positive values only during June in 2014, while it remains around zero during both 2007 and July of 2014.

Although the NAA and the BSA seem to share a common mechanism, giving rise to their similar maximum magnitudes, several relevant differences can be found between them. The most outstanding is probably the fact that the NAA is detected during the year with low solar activity at only around 18:00 LT in the evening but not during the night (a similar LT evolution of $D$ is observed in August, not shown in Figure 5). Hence, for the NAA, one can see an evening anomaly in 2007 with a magnitude two times smaller than in 2014, since the peak of $D$ is nearly 9% in July 2007 (also in August), while it is approximately 20% in June 2014. From Figure 5 (left bottom panel), we remark that, around 20:00 LT, the value of $D$ in the NAA region is approximately zero in July 2007 (also close to zero in June and August), so despite positive values of $D$ not being observed at 20:00 LT, as in 2014, there is still some hint of an anomaly since the EC at that time is still similar to midday values. This same situation is also found during the solar maximum in May, July, and August.

Another difference between the BSA and the NAA is that, in general, the EC enhancements in the BSA last for a longer LT period, normally between 2 and 4 h (see Figure 4), while clearly shorter enhancements are measured for the NAA (see Figure 5). Finally, no clear evidence of a morning peak in $D$ is seen in the NAA.

### 3.2. Analysis of Tidal and Planetary Wave Signatures in MSNAs

The contributions to the ionospheric EC from the different T/PW modeled by Equation (3) were analyzed after obtaining the best-fit model results. The individual fits in the magnetic apex latitude bins of 5° were averaged over the interval $[40°, 65°]$ in the northern (for the BSA and NAA) and southern (for the WSA) hemispheres. The results focus on the months in which the anomalies were maximal according to Table 1. Figure 6 shows the main T/PW components contributing to the formation of the WSA at different longitudes in January 2007 (left column) and January 2014 (right column). Figures 7–9 display the corresponding results for longitudes covering the BSA and the NAA for July 2007 (left columns) and June 2014 (right columns), respectively. In those two periods, no positive values of $D$ were obtained for longitudes around $-15°$, so in the case of the NAA, the results are only shown for the interval of longitude centered at $-45°$.

The relevant T/PW components represented in Figures 6–9 were chosen according to the following general criteria:

- The T/PW component must have a maximum around the LT period when an MSNA is observed, that is, when $D$ shows positive values according to the results presented in Section 3.1.
- The T/PW component must achieve a positive contribution, reaching near or over 10% of the sum of all components ($F(\lambda, t) - \overline{F}$ in Equation (3)), at least for a few hours during the LT period when the anomaly is observed.

Occasionally, a T/PW component that does not completely fulfill these criteria but is still relevant for a particular MNSA is also presented in the figures. We explicitly comment on this later in the text. The LT range of the horizontal axis for the figures was adapted for the relevant periods of the day when the corresponding anomalies are observed according to the results presented in Section 3.1. The vertical axis in the figures corresponds to the EC in TECU after subtracting the zonal time mean ($\overline{F}$), where only the positive contributions from the different waves are displayed. The sum of all T/PW components, $F(\lambda, t) - \overline{F}$, counting the positive and the negative contributions from all waves, is represented by the black dashed lines in Figures 6–9.

As can be seen from Figure 6, the main contributor to the WSA is the zero wave number diurnal tidal component, D0. This component depends only on the universal time, and hence, when looking westward in geographic longitude at 30° intervals, this wave appears to be shifted by two hours towards earlier LTs, as observed from the top to bottom panels in Figure 6. Following D0 in importance for the WSA, DW2 and SPW1 rank second, with similar contributions, which are about two times smaller than the amplitude of D0 for 2007 and nearly three times smaller than that for 2014. With a slightly smaller positive contribution to the WSA and particularly longitudinally to the east, the diurnal tide DE1 is next in importance. Finally, worth mentioning is the contribution from the so-called semidiurnal migrating tide SW2, $n = -s = 2$ in Equation (3), which peaks near midnight in 2007 and near 21:00 LT in 2014. Around those LTs and for a short period, SW2 provides a relevant contribution to the increase in nighttime EC causing the WSA, with a similar amplitude to the DE1 tidal wave in general. Recall that migrating tides ($n = -s$ in Equation (3)) only depend on LT, so in a given month and year, they have exactly the same LT profiles for all geographic longitudes (see individual columns in Figures 6 and 7).

Figure 6 also confirms that the main T/PW components producing the WSA in 2014 are the same as those in 2007, i.e., D0, SPW1, DW2, and DE1. For the LTs around midnight in 2007 and around 21:00 in 2014, SW2 is also included. However, from Figure 6, interestingly, the peaks of the main tidal waves contributing to the WSA, D0, and DW2 occur nearly two hours later in 2014 than in 2007, which agrees with the LT delay observed from parameter $D$ in the WSA, as noted in Section 3.1. On the other hand, when analyzing eastern longitudes in the WSA, Figure 6 (from the bottom to top panels) shows that the peak of D0 shifts towards later LTs and this fact, aided by a similar shift from DE1, supports the eastward drift with the LT of the WSA that can be seen in the black dashed lines, explaining the same drift in the parameter $D$ reported in Section 3.1. Moreover, the presence of an eastward drift is also supported by the fact that the peak in the planetary wave SPW1 occurs within (or close to) eastern longitudes of the WSA for both 2007 and 2014, which also helps to increase the anomaly in the eastern region around and after midnight LTs.

In the northern hemisphere, the importance of the semidiurnal migrating tide SW2 in producing the MSNAs is demonstrated by the results displayed in Figures 7–9), showing that SW2 is a major contributor to the production of both the BSA and the NAA during years with low and high solar activities. Worth remarking is that, contrary to the behavior shown by other tidal components, the peaks for SW2 during local summer in the northern hemisphere stay essentially at the same LTs, near 9:00 and 21:00, for the two years analyzed. In particular, we can conclude that the two EC peaks (during early morning and early night LTs) characterizing the BSA (see Figure 4) are clearly produced by the two peaks of the migrating semidiurnal tide SW2, shown in Figures 7 and 8, respectively.

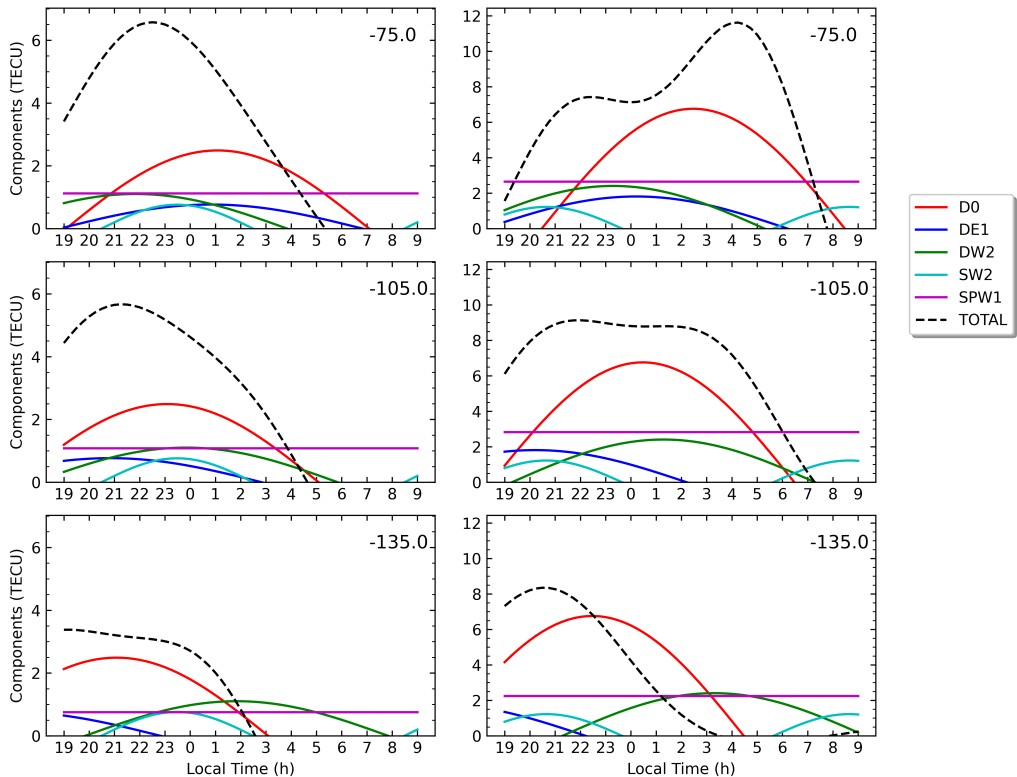

**Figure 6.** LT variation in the EC from the main T/PW components contributing to the WSA at different longitudes where the anomaly is located and after averaging over the magnetic apex latitude range [−65°, −40°]. The results correspond to January, the month showing the maximum EC enhancement. (**Left** column): 2007. (**Right** column): 2014. The different colored lines in the graphs represent different T/PW components according to the labels in the panel to the right of the graphs. The total represented by the black dashed line is the EC calculated after the sum of all T/PW components from Equation (3), including not only the positive but also the negative contributions from all waves.

According to Figure 7, for longitudes around 135°, the second most important contributor to nighttime BSA is DE1, which also peaks near 21:00 LT. For eastern longitudes, the contribution from DE1 decreases while that from DW2 increases, and these two tidal components contributed similarly to the BSA for the two years analyzed around 165°. Finally, we must note the presence of the westward terdiurnal and diurnal tides with wave number three, TW3 and DW3, which play secondary but still significant roles in maintaining the BSA, particularly in 2014, when they have clearly higher amplitudes. The terdiurnal migrating tide TW3 has a peak at the same LT when the BSA shows a nighttime peak in EC in 2014 (Figure 7, right column), contributing to nearly 25% of the sum of all T/PW components. In 2007, TW3 peaks one hour later than in 2014, just after 22:00 LT, and provides a contribution to the BSA slightly greater than 10% of the sum of all components. On the other hand, the westward diurnal component DW3 shows one peak at nearly 21:00 LT for the longitude sector centered in 165°, where it provides a similar contribution to the BSA as TW3 in 2014 (Figure 7, bottom row). A final remark concerns the role of the SPW1 in 2007, which contributed approximately 10% to the BSA for longitudes around 135° (Figure 7, top left panel). However, for the two years analyzed, the maximum for the planetary wave SPW1 during the northern summer period is not located in the region covered by the BSA. Probably for this reason, SPW1 does not significantly contribute to that anomaly at other longitudes or in 2014.

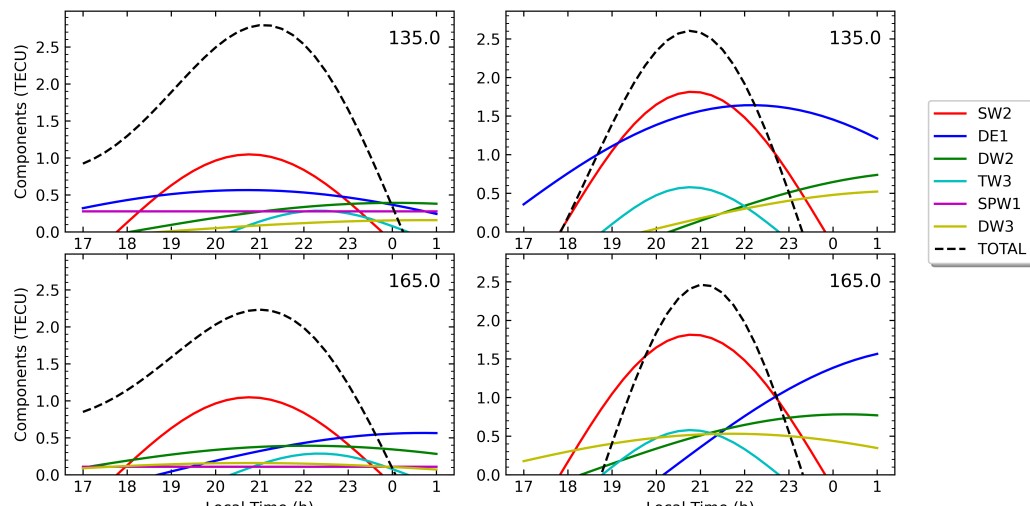

**Figure 7.** LT variation in the EC from the T/PW components after averaging over the magnetic apex latitude range [40°, 65°], which are the main contributors to the nighttime BSA at different longitudes where the anomaly is located. The results correspond to the months showing the maximum increase in EC. (**Left** column): July 2007. (**Right** column): June 2014. The legend to the right of the figure identifies the specific T/PW components displayed in the plots, and the black dashed line has the same meaning as in the previous figure.

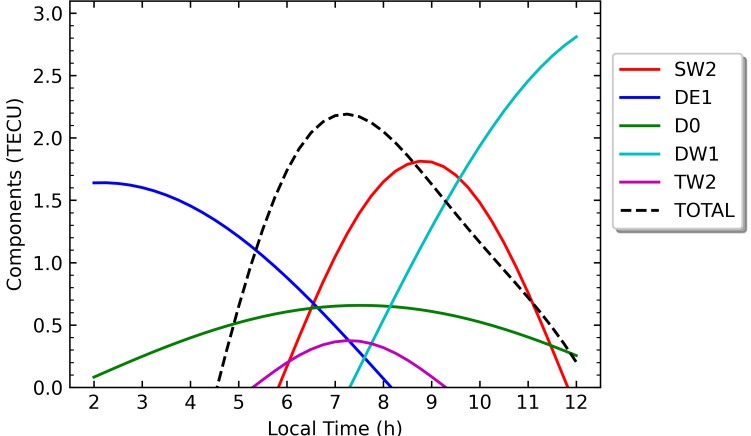

**Figure 8.** LT variation in the main T/PW components (after averaging over [40°, 65°] magnetic apex latitudes) contributing to the morning BSA. The *x*-axis is centered around the LTs for early morning, when the anomaly is maximal in June 2014 and for the longitude sector centered at 165°.

Figure 8 shows the analysis of the T/PW components specifically for the peak of EC enhancement in the BSA around 7:00–8:00 LT, which is clearly observed in the *D* values from Figure 4, particularly for the longitudes surrounding 165° during June 2014. According to the results presented in Figure 8 for the same period, this peak is produced essentially by the presence of the second peak of SW2 near 9:00 LT. However, the morning peak of the BSA has a significant difference with regard to the night peak concerning the T/PW components, providing a secondary contribution to that anomaly. One can see from Figure 8 that D0 replaces DE1 as the second-most important contributor to the morning BSA, and moreover, a new tidal component, TW2, appears as a third contributor that peaks between 7:00 and 8:00 LT. Instead, the peaks of DE1 and other T/PW components, including those playing secondary roles in the nighttime BSA, are very far from the morning LTs to contribute to the BSA during that period.

The role played by the diurnal migrating tide DW1 in northern MSNAs deserves a special remark. DW1 is the tidal wave with the largest amplitude among all T/PW components in the magnetic apex latitude interval $[40°, 65°]$. Its amplitude is approximately 70% larger than the amplitudes of SW2 and DE1, but the peak value of the DW1 wave happens near 13:00 LT in June 2014 and a little before 15:00 LT in July 2007, clearly during the midday period. Hence, DW1 nearly vanished at 19:00 LT in 2014 and a little before 21:00 LT in 2007, being negative during the night, and consequently, it does not significantly contribute to the nighttime BSA. Instead, DW1 plays a key role counteracting the other T/PW components during the night and preventing the BSA from being observed after 22:00 LT, when the DW1 wave is largely negative. Recall that the values of parameter *D* in Figure 4 decrease quickly to zero after 22:00 LT. On the other hand, as can be observed in Figure 8, during the morning, DW1 approaches zero near 7:00 LT, but for later LTs, this wave is positive and increases quickly due to its large amplitude. At 8:00, the EC contribution from DW1 is slightly greater than the one from TW2 and, soon after, is similar to the contribution from D0. Thus, clearly, DW1, together with TW2 and D0, has a relevant secondary role in the development of the morning BSA.

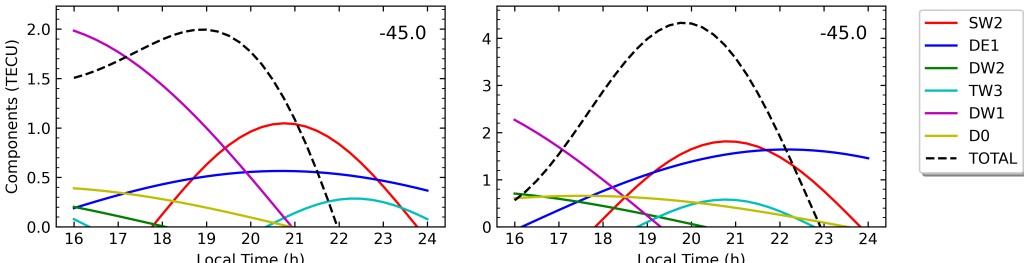

**Figure 9.** Same as Figure 7 but for the main T/PW components contributing to the NAA in longitudes around $-45°$ and for a LT range around 20:00.

Finally, from Figure 9 (right panel), one can observe that the NAA is formed in 2014 from the superposition around 21:00 LT of the peaks of SW2 and DE1 and, secondarily, of the peak of TW3, which is the same as in the case of the nighttime BSA. However, in the NAA, the proximity of the peak from D0 replaced the contributions from DW2 and DW3 that were observed in 2014 for the nighttime BSA. In fact, neither previous component contributes significantly to the NAA in the two years analyzed. On the other hand, comparing the left and right panels of Figure 9, one can observe (black dashed lines) that the NAA appears to be shifted towards earlier LTs in the evening, around 18:00–19:00, in 2007. In particular, compared with 2014, in 2007, the amplitude of DE1 clearly decreases compared with the amplitude of SW2; moreover, the peak of TW3 is delayed until after 22:00 LT. Those two facts contribute to the NAA being absent around 20:00–21:00 LT in 2007. However, the main factor that explains the increase in EC during 18:00–19:00 LT in 2007 is clearly the large amplitude of the DW1 component (Figure 9, left panel). As mentioned before, the DW1 wave peaks close to 15:00 LT in 2007, having the largest amplitude among all T/PW components. For that reason, the contribution to the NAA from DW1 between 18:00 and 19:00 LT is certainly large in that year and, in fact, is even greater than the EC contributed by SW2 and DE1 at those LTs. Thus, DW1 provides the main positive contribution to the evening NAA in 2007, explaining the peak that was reported in Figure 5 (left column). Nevertheless, the proximity in LT of the peaks from SW2 and DE1 also helps to support the presence of this evening anomaly. On the contrary, as noted before for the BSA, during 2014, DW1 nearly vanishes at 19:00 LT (Figure 9 right panel) and, consequently, in that year, cannot contribute to the NAA that is observed later at 20:00 LT.

## 4. Discussion

In general, the results achieved in this study on the observational signatures of MSNAs agree with that from previous research for periods with minimal or low solar activity. Moreover, our study extends the results to the last solar maximum year. However, our $EC_{ion}$ data sample shows that, in 2007, the NAA is detected essentially in the evening. Instead, this anomaly is observed in the same year at 20:00 and 22:00 LT when using ED values at an altitude of 300 km obtained from ROs [1]. Thus, the integrated ionospheric EC probably does not provide sufficient spatial and temporal resolution to predict the NAA, because this anomaly is too localized in altitude (and probably in geographic area, see Figure 2) to be observed by means of $EC_{ion}$ when averaged over the magnetic apex latitude range used in this study. In fact, according to our observational results in July (Figure 5, bottom row), the increases in ionospheric EC in the NAA are close to zero in 2007 at 20:00 LT, which indicates the presence of a weak MSNA. A few hours before, at 18:00 LT, the values of $D$ in 2007 become positive, but never exceed 10%. Finally, our results in 2014 clearly show the presence of the NAA at the start of nighttime, with a similar magnitude to that of the BSA. Then, our conclusion from the observed $EC_{ion}$ is that the NAA is slightly more prominent during the solar maximum than during a period of low solar activity.

Concerning the best-fit model results from the frequency-wave number analysis, the most striking difference found in our study with regard to previous studies is the major role played by the peaks of the semidiurnal migrating tide, SW2, in the production and maintenance of the MSNAs in the northern hemisphere, not only during the night between 20:00 and 22:00 LT, but also during the morning, with a peak in the BSA at 8:00 LT. The importance of SW2 for the formation of the BSA and the NAA is a result not previously reported, to our knowledge, and this could provide new insights for future theoretical analysis. Additionally, SW2 has been found to peak at nighttime in the WSA region, but its amplitude is smaller compared with the other components producing that anomaly. Thus, although playing a secondary role in the formation of the southern MSNA, its contribution cannot be neglected. Previous studies [3,11] based on data from the solar minimum around 2009 concluded that increases in EC in summer nighttime observed in the northern hemisphere are produced mainly by DE1 and DW2, but nothing has been said about SW2, which according to the results of this study, is the most relevant component producing the MSNAs in the northern hemisphere. Additionally, these studies point out that D0 also has some secondary role in producing these anomalies. From Figure 9, we show some evidence that this could be the case for the NAA in 2014, when D0 appears to play a very minor role, but not in 2007, near the solar minimum. Additionally, for the nighttime BSA, the secondary role is played by other tidal components, such as DW3 and TW3, while D0 has a negative contribution during the night. Instead, D0 peaks between 7:00 and 8:00 LT (Figure 8) and, consequently, certainly provides some contribution to the formation of the early morning peak in the BSA, although not at all for the nighttime peak around 21:00 LT. Most of the T/PW components that appear to contribute to the northern MSNAs were shown to also be present during June 2007 in the vertical component of neutral wind along the magnetic field lines at an altitude of 200 km and at the geomagnetic latitude 45° [21], according to the Thermosphere–Ionosphere Electrodynamics General Circulation Model (TIEGCM) results. This indicates a direct relationship between vertical winds along geomagnetic field lines in the thermosphere and electron density in the MSNAs, as pointed out in previous studies [1,2,13,14]. Since the T/PW signatures in the northern MSNAs in 2014 are similar to those observed in 2007, this suggests that the same relationship also exists during the solar maximum period.

Migrating tides (DW1, SW2, and TW3) have been demonstrated to be directly related to the formation of equatorial ionization anomaly crests [19,32], but not much attention has been paid to their role in the formation of MSNAs specifically. Apart from the importance of SW2 for the MSNAs shown in our results, the role of DW1 in the evening NAA in 2007 (Figure 9, left panel) and in the maintenance of the morning BSA (Figure 8) was confirmed to be relevant. Finally, the migrating terdiurnal tide TW3 was shown to be among the

tidal components supporting the BSA and the NAA during the night in 2014 and, to a minor extent, in 2007. The origin of the LT variation in ionospheric migrating tides was explored by means of TIEGCM simulations in [19]. Although that study focused primarily on low-latitude regions, the results were presented in the geomagnetic latitude range from −50° to 50°. Thus, from [19], the role of upward propagating migrating tidal components from the neutral mesosphere and lower thermosphere can be assessed in the corresponding midlatitude migrating tides that are observed to contribute to the northern MSNAs. The results from [19] indicate that the amplitude of SW2 has a small peak during July 2007 in magnetic latitudes around 45°. A qualitative estimate of the peak from Figure 3 of [19] yields a value close to 1 TECU, which is consistent with our results in July 2007 (see the SW2 amplitudes observed in Figures 7 and 9, left columns). Later, in Figure 6, from [19], the reconstructed TEC diurnal variations from F3C data show that the SW2 migrating tide has two clear peaks at 45° magnetic latitude in the summer solstice period and the peaks occur at nearly the same LTs as those observed in our Figures 7–9. After performing different numerical runs of the TIEGCM, the authors of [19] concluded that those midlatitude enhancements in SW2 are produced in situ and are not the result of forces from migrating tides propagating upward from the mesosphere and lower thermosphere regions.

Previous numerical studies [19,21] have not considered geomagnetic latitudes greater than 45°–50° in their studies of the T/PW components of northern MSNAs. Thus, exploring the existence of the SW2 peak and other non-migrating tides that secondarily contribute to the northern MSNAs at greater latitudes can be interesting. Toward this aim, we performed a specific analysis in the upper range of magnetic apex latitudes [50°, 65°], re-calculating the T/PW components for July 2007 and June 2014 after averaging them only over that interval. The results are illustrated in Figure 10, which can be compared with the previous Figure 7 (top row) and Figure 9, where the T/PW components were averaged over the full interval [40°, 65°]. In Figure 10, one can observe that SW2 is still the main tidal component contributing to the BSA (top row of the figure) and to the NAA (bottom row of the figure) in the two years analyzed. In fact, only slight changes in the peak location and amplitude can be seen, not only for SW2 but also for the other tidal components contributing to the northern MSNAs, as can be verified by comparing Figure 10 with Figures 7 and 9. Hence, one can conclude that the relevance of the migrating semidiurnal tide SW2 and the other secondary T/PW components for the MSNAs in the northern hemisphere is not restricted to the interval with the lowest magnetic apex latitudes, from 40° to 50°. Thus, we can reasonably assume that the same physical mechanisms argued in [19,21] to explain the origin of the tidal features in the ionospheric MSNAs at latitudes 45° are also valid for higher latitudes. Dedicated numerical simulations should confirm this conclusion.

In the case of the WSA, this study confirmed the findings from previous research during years of low solar activity about the major T/PW components contributing to that anomaly [3,11,21]. The mechanisms behind the generation of those components has been explored by means of numerical simulations performed with the Ground-to-Topside Model of Atmosphere and Ionosphere for Aeronomy (GAIA) general circulation model [11]. From that study, the authors concluded that the ionospheric tides can be generated due to a combination of two mechanisms: in situ photoionization and plasma transport along magnetic field lines. In particular, this conclusion applies to D0, the main tidal component contributing to the WSA. If the contribution to the WSA by some T/PW component was dependent on the in situ photoionization, one should expect some dependence of the observed magnitude of that component with solar activity. This seems to be the case for D0, as observed from Figure 6, where the amplitude of D0 is nearly two times larger than the other secondary components in 2007 but about three times greater in 2014, probably due to more effective in situ photionization during the year of solar maximum.

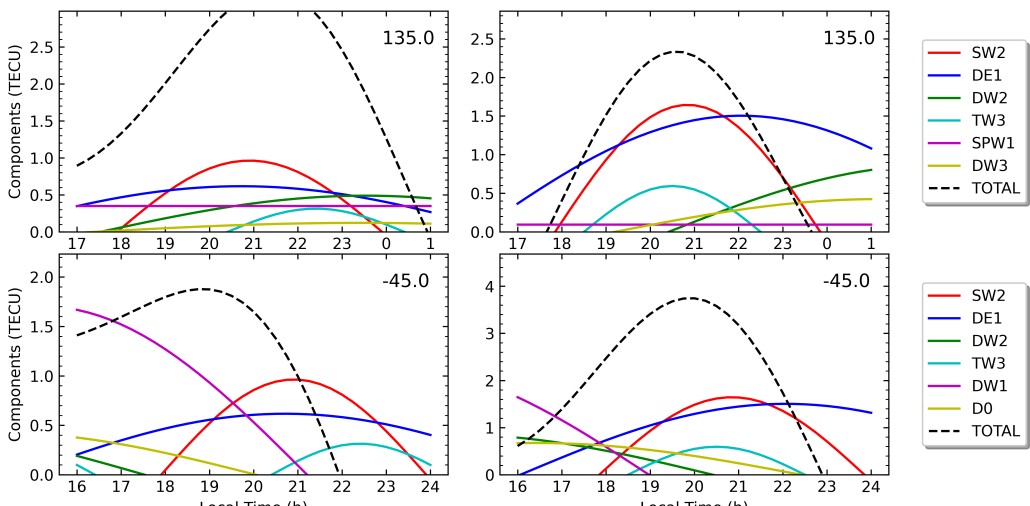

**Figure 10.** The LT variation in the main T/PW components contributing to the BSA at longitude 135° (**top** row) and to the NAA at longitude −45° (**bottom** row) after averaging over the restricted interval [50°, 65°] of magnetic apex latitudes. (**Left** column): July 2007. (**Right** column): June 2014.

Despite the similarities to the conclusions of previous studies in the case of the WSA, a feature observed from the frequency–wave number analysis in this study is the relative importance attributed to the stationary planetary wave SPW1 in the formation of the WSA. According to our results, SPW1 provides a more significant contribution to the WSA than previously reported in [11]. We found that, in both 2007 and 2014, the peak of the planetary wave SPW1 occurs well within the region covered by the WSA, close to eastern longitudes according to our model fit to the different bins of magnetic apex latitude. However, in the study by [11], they assigned a minor role to this planetary wave component, which they estimated to provide only around 5% of the total contribution to the WSA in the solar minimum period. Instead, according to our results, SPW1 contributes to the formation and maintenance of the WSA in 2007 and 2014, providing between 20 and 25% of the total contribution, at a similar level to DW2, which has an amplitude similar to the stationary electron content contributed by SPW1. The importance of the SPW1 for the formation of the WSA during a deep solar minimum in 2009 was highlighted by [3] after a one-dimensional Fourier analysis of observed tidal signatures from non-migrating tides and by model simulations in [21]. Hence, this study confirms these previous results and extends them to 2014, a period with high solar activity. Together with D0, SPW1 has been shown to have the largest amplitude in the vertical wind along geomagnetic field lines in the thermosphere [21] for a magnetic latitude of −45° and at an altitude of 200 km. However, according to GAIA simulations [11], whether the contribution to the WSA from the SPW1 is generated by neutral winds produced by in situ photoionization or is related to plasma transport by effective winds is unclear. More dedicated numerical experiments are necessary to clarify this point.

It can be argued that some of the differences found in this study compared with those of previous research, particularly concerning the T/PW components having a secondary or minor contribution to the MSNAs, might be attributed to differences in the averaging strategy employed to evaluate the tidal signatures from the model fits. For example, while in this work, we focused on fits for each month, the study of the WSA by [11] used to calculate their results a 40-day averaging window, stepped daily, from mid-November 2008 until mid-February 2009. Hence, we explored if the averaging procedure used to calculate the T/PW components may have some impact on the results. Toward this aim, we repeated the frequency–wave number analysis and calculated the tidal signatures following the same averaging strategy used in [11] for the same period used in that paper, but for 2007 instead of 2009. Both years belong to the same solar minimum period between solar cycles 23 and

24, and the solar activity was somewhat lower in 2009 than in 2007, but was not too different. The results after the new calculations are illustrated in Figure 11 for two longitudes in the WSA region. The most relevant conclusion is that the T/PW components with the largest amplitudes in the nighttime period are not changed with regard to the ones presented in Figure 6 (left column). In particular, we confirm the importance of the planetary wave SPW1 for the formation of the WSA, with D0 being the main tidal component from which the WSA originates, while DE1 and DW2 make secondary contributions, which are lower but not very far from that of SPW1.

However, by comparing Figure 11 with Figure 6, some differences can be noticed. Clearly, the new averaging procedure smoothed the results, since the peak of the sum of all T/PW (black dashed line) and the amplitudes of the different components shown in Figure 11 are smaller than the values seen in Figure 6. The observed smoothing is not surprising, since averaging with a 40-day sliding window over a period that covers several months, the strategy used by [11], should have this effect compared with the approach based on monthly averages, followed in this study. Even though the reduction in the amplitudes of the T/PW components shown in Figure 11 is small for D0 and SPW1 when compared with Figure 6, the amplitude of SW2 is nearly three times smaller and that for DW2 approximately two times smaller. This may explain why the contribution of SW2 to the WSA was not highlighted by [11]. Despite this, one can observe in Figure 11 that SW2 still shows a peak approximately when the WSA anomaly maximizes the increase in EC around 22:00 LT. Hence, except for some small variations in the amplitudes and LTs of the peaks of the tidal components, the results confirm that the averaging procedure does not have a relevant impact on the conclusions about the main T/PW components contributing to the WSA. This conclusion supports that the differences in our results compared with previous studies can be attributed to the use of the $EC_{ion}$ data and, consequently, to the use of a more accurate methodology for retrieving ED profiles and for calculating specific ECs from the ionosphere region.

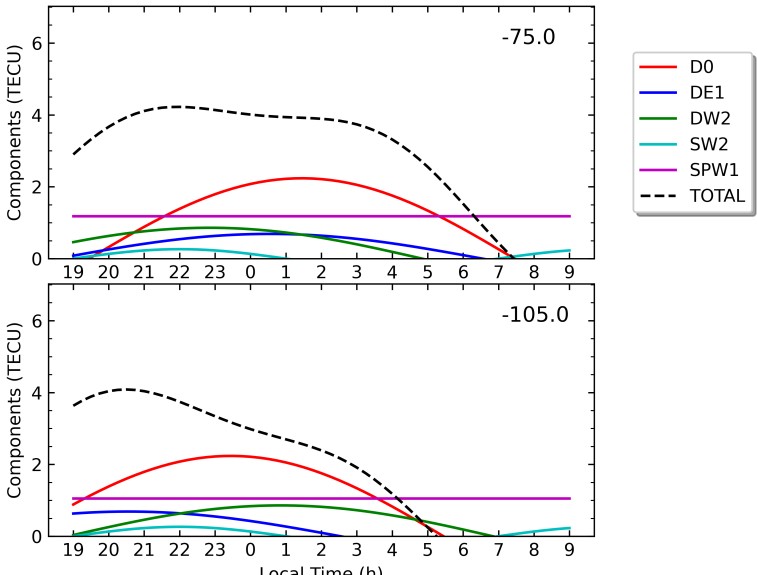

**Figure 11.** LT variation in the main T/PW components contributing to the WSA in the magnetic apex midlatitude range $[-65°, -40°]$, after following the alternative procedure of averaging the model fits over a daily stepped moving window covering an interval of 40 days, for days 315–365 and days 1–45 in 2007 and for the longitudes indicated in the top right corner of the figures.

## 5. Conclusions

We investigated the increase in summer nighttime ionospheric EC at midlatitudes from observations and T/PW features, comparing two years of extreme solar activity: 2007 and 2014. For this study, a new quantitative variable was used: the EC from the ionosphere calculated after excluding the plasmaspheric contribution to the TEC. The data set was derived following an accurate methodology used to retrieve ED profiles from ROs, which was developed and verified in several previous studies. Our investigation focused on the comparison between different MSNAs in two years with low and high solar activities, looking specifically at the temporal and spatial characteristics of the observed anomalies and, subsequently, exploring the T/PW components contributing to MSNAs.

Our main conclusion is that the use of the ionospheric EC provides relevant information about observational aspects of the MSNAs and their main T/PW signatures. In particular, the results from this empirical study are of interest in furnishing the constraints and characteristics of the MSNAs for subsequent modeling and theoretical analyses to be performed in future studies.

The main findings of our investigation can be summarized as follows:

- Solar activity does not significantly affect the maximum relative magnitude of MSNAs measured by the maximum increase in EC at night compared with the midday period $D_{max}$. The major T/PW components producing MSNAs also do not differ between low and maximum solar activity periods. This fact points to a mechanism related mainly to internal atmospheric processes that act similarly during all periods of the solar cycle in creating this type of anomaly. In particular, this suggests that plasma transport by effective winds that may propagate from the lower thermosphere is a potential mechanism that generates the ionospheric tides contributing to the MSNAs.
- However, some particular behaviors of the MSNAs seem to be influenced by solar activity. The main evidence of this is the different LT evolution of the anomalies between 2007 and 2014 for fixed longitudes, which is reflected in the different months and LTs for $D_{max}$ values in the northern MSNAs and the different longitudes in the case of the WSA. The observed differences can be attributed to the locations of the maximum values of the T/PW components in the LT/longitude plane and the fact that non-migrating tides producing the MSNAs reach their maximum values 2–3 h later in 2014 than in 2007. The dependence of those features on the solar cycle indicates that a mechanism, such as in situ photoionization, which is affected by solar activity, may also be responsible for generating the neutral winds that contribute to the ionospheric T/PW signatures observed in the MSNAs.
- MSNAs are confirmed to be detectable in three different regions. The WSA in the southern hemisphere shows increases in nighttime EC that are significantly larger, more than three times as large, than those in the other two MSNA regions in the northern hemisphere. The WSA is observed during the night from 18:00 to 8:00 LT and during several months surrounding the local summer period, particularly in the eastern longitudes between $-60°$ and $-90°$. On the other hand, considering increases in evening and nighttime EC, the NAA and the BSA are not very different in magnitude, although the NAA appears to be slightly weaker in 2007 compared with 2014. The BSA is observed during early night, 20:00–22:00 LT in the two years, and during the early morning, 6:00–8:00 LT, in 2014, being also hinted at during the morning in 2007. The NAA is observed during the evening around 18:00 LT in 2007 and during the early night period around 20:00 LT in 2014.
- The main tidal component that appears to give rise to the WSA is D0. Next, SPW1 and DW2 have a secondary but still relevant contribution at similar levels, followed by DE1. The BSA and NAA are clearly mainly supported by the migrating tide SW2 in combination with DE1, but in the case of the morning BSA, D0 replaces DE1 as having a secondary role. Another migrating tide, DW1, has been shown to provide a supporting contribution to the morning BSA in 2014 and to the evening NAA during

2007. Finally, a third migrating tide, TW3, appears to contribute to both the nighttime BSA and the NAA, particularly during the solar maximum period.

- For the two years analyzed, peaks in SW2 are also observed at the LTs when the WSA is observed. Thus, this semidiurnal migrating tide also has some relevance in maintaining the WSA at certain hours of the night, although due to the use of a larger smoothing, its relevance was probably not as evident in previous studies [11]. Moreover, the importance of the SW2 migrating tide for the MSNAs in the northern hemisphere was shown to be effective in the specific interval of magnetic apex latitudes $[50°, 65°]$, and not only for the lower range of magnetic apex midlatitudes $[40°, 50°]$, as indicated by previous results [19].

**Author Contributions:** Conceptualization, formal analysis, methodology, and investigation, G.G.-C., Y.Y., J.M.J., A.R.-G. and J.S.; resources and software, Y.Y. and Y.S.; supervision, G.G.-C., A.R.-G. and J.M.J.; validation, A.R.-G. and J.S.; writing—original draft preparation, G.G.-C. and Y.Y.; writing—review and editing, J.M.J., A.R.-G., J.S. and Y.S.; project administration and funding acquisition, G.G.-C. and J.M.J. All authors have read and agreed to the published version of the manuscript.

**Funding:** This research and the APC were funded by the Spanish MCIN and by the European Comunity FEDER through the RETOS project number RTI 2018-094295-B-I00. Yu Yin acknowledges financial support from the China Scholarship Council through grant no. 202006020025.

**Institutional Review Board Statement:** Not applicable.

**Informed Consent Statement:** Not applicable.

**Data Availability Statement:** The data set and results from this study are available upon reasonable request to the corresponding author.

**Acknowledgments:** The authors acknowledge the use of public data and products from the International GNSS Service and from the COSMIC Data Analysis and Archive Center.

**Conflicts of Interest:** The authors declare no conflict of interest. The funders had no role in the design of the study; in the collection, analyses, or interpretation of data; in the writing of the manuscript; or in the decision to publish the results.

## Abbreviations

The following abbreviations are used in this manuscript:

| | |
|---|---|
| BSA | Bering Sea Anomaly |
| EC | Electron content |
| ED | Electron density |
| F3C | Formosat-3/Constellation Observing System for Meteorology, Ionosphere, and Climate |
| GAIA | Ground-to-Topside Model of Atmosphere and Ionosphere for Aeronomy |
| GIM | Global ionospheric map |
| GNSS | Global navigation satellite system |
| LT | Local time |
| MSNA | Midlatitude summer nighttime anomaly |
| NAA | North Atlantic Anomaly |
| RO | Radio occultation |
| SM | Separability method |
| TEC | Total electron content |
| TIEGCM | Thermosphere–Ionosphere Electrodynamics General Circulation Model |
| TIP | Topside ionosphere plus bottom-side plasmasphere |
| T/PW | Tidal and planetary wave |
| WSA | Weddell Sea Anomaly |

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
