# Peer review of "Summer Nighttime Anomalies of Ionospheric Electron Content at Midlatitudes: Comparing Years of Low and High Solar Activities Using Observations and Tidal/Planetary Wave Features"

_remotesensing, doi:10.3390/rs14051237_

Round 1

Reviewer 1 Report

This paper provides a detailed analysis of the planatary wave and tidal signatue in ionosphere TEC based on COSMIC observation of the Middle-latitude summer nighttime anomaly, including wedell sea anomaly and north atlantic anomaly. Unlike previous studies that direct using the satellite measurements, this study proposed the potential harm of direct using, but try to separate the measurements into plasmasphere and ionosphere electron content. This is a creative aspect. Furthermore, this study picked one solar-minimum year and one solar-maximum year for comparison, which has not been carried out before for those studies focusing on MSNA. The paper is well organized, and the results are in detail. The only disadvantage is that many sentences are orally written, and do not obey the common english grammar. In all, I recommend this paper to have a minor revision before suitable for publication.

Line 26 I think the author’s meaning is the nighttime electron density is larger than the one during local noon. Please correct into ‘local noon

Line 36 replace ‘along’ with ‘in’’

Line 42 reference 14 Zhong et al., 2017 is not studying the WSA, they only metioned WSA in their results. So the author shall delete this reference

Line 46 replace ‘onset’ with ‘launching’

Line 209-212 the author shall give a certain value, which is easy to obtain from the data, not just say ‘smaller

Figure 1, it would be better for author to add the corresponding error bar onto these results, which shall be more convincible

Line 218-219 replace them with ‘In the present study, based on the observational data sample of ECion and the method from [1]’’

Line 223-226 from ‘Then, by means of XXX’    This sentence is too long and orally-written. Please use two separate sentences

Line 232 again, orally written sentence. Correct into ‘After obtaining the fitting results’

Line 244 replace ‘have used’ with ‘use’

Line 351-353, again, orally-written sentence.

Line 454-456 again, orally-written sentence.

Reviewer 2 Report

Midlatitude summer nighttime anomaly (MSNA)is a phenomenon that the electron density in the ionosphere is larger during nighttime than during daytime. Declination angle of the geomagnetic field has been considered to be responsible for generation of MSNA. At the location of the westward (eastward) declination, the eastward thermospheric neutral winds push the F-region plasma along the magnetic field line to higher altitude, where the recombination rate is small, so that the F-region plasma keep high electron density. As another effect, poleward shift of the magnetic equator with respect to the geographical equator contribute to the generation of MSNA. For this effect, the meridional wind along the magnetic field line plays an important role in the MSNA generation (Lin et al., 2009). 
In this paper, however, the well-known effects, described above, are not argued. The authors need to explain mechanism how the tides and planetary waves studied in this paper are related to the above effects and the mechanism generating MSNA.

The authors obtained tidal and planetary wave components from the Fourier analysis of the electron density data instead of the data of neutral atmosphere. In general, tidal and planetary waves are phenomena of the neutral atmosphere. The authors need to describe how the tidal and planetary wave component of the electron density is related to the  tidal and planetary waves in the neutral atmosphere.

Lin, C. H., Liu, J. Y., Cheng, C. Z., Chen, C. H., Liu, C. H., Wang, W., Burns, A. G., and Lei, J. (2009), Three-dimensional ionospheric electron density structure of the Weddell Sea Anomaly, J. Geophys. Res., 114, A02312, doi:10.1029/2008JA013455.

Reviewer 3 Report

Comments to the manuscript remotesensing-1610333

Summer Nighttime Anomalies of Ionospheric Electron Content at Midlatitudes: Comparing between years of Low and High Solar Activity using Observations and Tidal/Planetary Wave Features

by Yu Yin, Guillermo González-Casado, Adrià Rovira-Garcia, Jose Miguel Juan, Jaume Sanz and Yixie Shao

The paper is devoted to the study of the Midlatitude Summer Nighttime Anomaly (MSNA) based on F3C satellite data. The MSNA phenomenon was considered for three zones of the Northern and Southern hemispheres and it turned out that in all cases the main role is played by tidal and planetary waves, only the contributions of various components are different. Moreover, the results for high solar activity were close to the features found during the period of low activity. Thus, for the first time, the role of the SW2 component in the production and sustenance of the MSNAs in the North hemisphere was revealed. The advantage of the study is the use of the author's method of dividing TEC into ionospheric and plasmaspheric parts.

Some remarks are as follows.

Comments

  1. Line 29: the use of arbitrary numbering of references is not entirely clear.
  2. Lines 208-209: it is desirable to supplement Figure 1 with the same Figure for July, so that the difference of 80-100% for the Southern hemisphere is clear.
  3. Lines 246-252: It is not entirely clear why SPW refers to planetary waves if the symbol S stands for the semidiurnal tidal component.
  4. Line 454, Figure 6: why the sum of all components (black dashed lines) is zero, even though individual components may exist.

Round 2

Reviewer 2 Report

Incorporating the review comments, the manuscript has been revised. The current version is acceptable for publication.